# Molecular Basis for Paradoxical Activities of Polymorphonuclear Neutrophils in Inflammation/Anti-Inflammation, Bactericide/Autoimmunity, Pro-Cancer/Anticancer, and Antiviral Infection/SARS-CoV-II-Induced Immunothrombotic Dysregulation

**DOI:** 10.3390/biomedicines10040773

**Published:** 2022-03-25

**Authors:** Tsai-Hung Wu, Song-Chou Hsieh, Tsu-Hao Li, Cheng-Hsun Lu, Hsien-Tzung Liao, Chieh-Yu Shen, Ko-Jen Li, Cheng-Han Wu, Yu-Min Kuo, Chang-Youh Tsai, Chia-Li Yu

**Affiliations:** 1Division of Nephrology, Taipei Veterans General Hospital, National Yang-Ming Chiao-Tung University, Taipei 11217, Taiwan; thwu@vghtpe.gov.tw; 2Department of Internal Medicine, National Taiwan University Hospital, Taipei 10002, Taiwan; hsiehsc@ntu.edu.tw (S.-C.H.); b89401085@ntu.edu.tw (C.-H.L.); tsichhl@gmail.com (C.-Y.S.); dtmed170@gmail.com (K.-J.L.); chenghanwu@ntu.edu.tw (C.-H.W.); 543goole@gmail.com (Y.-M.K.); 3Division of Allergy, Immunology and Rheumatology, Shin Kong Wu Ho Shi Hospital, Taipei 11101, Taiwan; pearharry@yahoo.com.tw; 4Institute of Clinical Medicine, National Yang-Ming Chiao-Tung University, Taipei 11217, Taiwan; 5Institute of Clinical Medicine, College of Medicine, National Taiwan University, Taipei 10002, Taiwan; 6Division of Allergy, Immunology and Rheumatology, Taipei Veterans General Hospital, National Yang-Ming Chiao-Tung University, Taipei 11217, Taiwan; darryliao@yahoo.com.tw

**Keywords:** polymorphonuclear neutrophil, mitogen-induced cell-mediated cytotoxicity, antibody-dependent cell-mediated cytotoxicity, neutrophil extracellular traps, ectosomes, exosomes, trogocytosis, SARS-CoV-2 pandemic, immune homeostasis, immunothrombosis

## Abstract

Polymorphonuclear neutrophils (PMNs) are the most abundant white blood cells in the circulation. These cells act as the fast and powerful defenders against environmental pathogenic microbes to protect the body. In addition, these innate inflammatory cells can produce a number of cytokines/chemokines/growth factors for actively participating in the immune network and immune homeostasis. Many novel biological functions including mitogen-induced cell-mediated cytotoxicity (MICC) and antibody-dependent cell-mediated cytotoxicity (ADCC), exocytosis of microvesicles (ectosomes and exosomes), trogocytosis (plasma membrane exchange) and release of neutrophil extracellular traps (NETs) have been successively discovered. Furthermore, recent investigations unveiled that PMNs act as a double-edged sword to exhibit paradoxical activities on pro-inflammation/anti-inflammation, antibacteria/autoimmunity, pro-cancer/anticancer, antiviral infection/COVID-19-induced immunothrombotic dysregulation. The NETs released from PMNs are believed to play a pivotal role in these paradoxical activities, especially in the cytokine storm and immunothrombotic dysregulation in the recent SARS-CoV-2 pandemic. In this review, we would like to discuss in detail the molecular basis for these strange activities of PMNs.

## 1. Introduction

Human polymorphonuclear neutrophils (PMNs) are the most abundant leukocytes in the circulation, carrying out phagocytosis and killing of the invading pathogens. Traditionally, PMNs are considered as a terminally differentiated and homogeneous population with a short lifespan and low transcriptional capacity [1,2]. Functionally, these cells are oriented to as the first-line responders and powerful guardians of the body’s defense against invaders. They protect the body by phagocytosis, intra-phagolysosomal killing, release of proteolytic enzymes through degranulation, production of reactive oxygen species (ROS), and formation of neutrophil extracellular traps (NETs). Recently, many investigators have revealed the heterogeneity [3] and various novel biological functions of PMNs. This heterogeneity has been particularly identified in certain disease entities [3,4]. For instance, low-density granulocytes (LDGs) in SLE [5,6,7,8,9], myeloid-derived suppressor cells (MDSCs) in specific inflammatory diseases [10], and the N1 neutrophils (N1) with anticancer capacity have been found [11,12]. Besides, in the last decade, scientists have further understood that PMNs are much more complex cells, participating in modulating adaptive immune responses, anti-inflammatory, antiviral, pro-cancerous and anticancerous immunity, as well as an inducer of severe COVID-19 (SARS-CoV-II) complications including hyperinflammation/necroinflammation of the lung, cytokine storm, immunothrombosis, and cardiovascular disease (CVD). These findings have suggested that PMNs not only work at the crossroads of innate and adaptive immunity [13], but act as a double-edged sword with various paradoxical activities. They can either protect the body against infections and cancers, or become an unpredictable rebeldom in the COVID-19 pandemic. We are going to discuss in detail the PMNs regarding their granulopoietic regulation, novel biological/immunological functions, and paradoxical activities in the present review. 

## 2. Regulation of Granulopoiesis, Response to Environmental Factors, and Destinies of PMNs in the Body

PMNs are professional phagocytes growing and differentiating in the bone marrow. Autophagy machineries are highly conserved in these cells [14]. Autophagy is an intracellular homeostatic mechanism of eukaryotic cells essential for the cellular response to starvation or other types of stress such as hypoxia/oxidative burst, DNA damage and infections [15]. The importance of autophagy for efficient differentiation has been demonstrated in lymphocytes [16,17,18], monocytes [19], dendritic cells [20], as well as reticulocytes [21,22], but has been overlooked in neutrophils. However, autophagic regulation is crucial for the effector functions of PMNs, as shown in the following aspects.

### 2.1. Regulatory Roles of Autophagy in Neutrophil Effector Functions

Rozman et al. [23] disclosed that autophagy is not essential for neutrophil granulopoiesis. Instead, the autophagic activity correlates inversely to the rate of neutrophil differentiation. Thus, a reciprocal relationship between autophagy and PMN differentiation is present, which is modulated by the *p*38-mTORC1. Nevertheless, autophagy is required for many PMN-mediated effector functions such as granule formation, degranulation, release of neutrophil extracellular traps (NETs), cytokine production, bacterial killing and/or inflammation control [24,25,26,27,28]. Figure 1 depicts the regulatory roles of autophagy in many immune cell functions including those of PMNs but not granulopoiesis.

### 2.2. Rapid Sensing and Effective Response of PMNs to Environmental Factors

PMNs can detect different pathogens by using an array of surface-expressed innate immune receptors. Toll-like receptors (TLRs) can sense and bind the pathogen-associated molecular pattern (PAMPs) and damage-associated molecular patterns (DAMPs) from variant origins [29]. Complement receptors (CRs) can identify and attach the complement-fixed immune complexes [30,31]. Low- and high-affinity IgG Fc receptors can recognize and grasp antibody-attached antigens [32]. Thereby, PMNs act as powerful effectors for TLR-, complement- and antibody-mediated inflammation. In addition, PMNs also play a pivotal role in the resolution of tissue inflammation via secretion of lipid anti-inflammatory molecules, resolvins, to inhibit inflammatory reactions [33]. In short conclusion, PMNs can be affected by the environmental factors to which they are exposed and partake in coordinating inflammation and/or anti-inflammation processes for immune adaptation and homeostasis, as shown in Figure 2.

### 2.3. Factors Influencing the Destinies of PMNs

Under physiological conditions, the mature circulating PMNs have a very short half-life of no more than one day in vivo [34,35]. However, Pillay et al. [36] conducted an in vivo incubation of human PMNs with ^2^H_2_O and revealed that the lifespan of normal PMNs is around 5.4 days. Ordinarily, the lifespan of PMNs can be lengthened during the course of inflammatory response by signals in the inflammatory milieu [37]. In general, these cells constitutively undergo spontaneous apoptosis, but their short-time survival can be extended by certain growth factors, proinflammatory cytokines or bacterial products including G-CSF, GM-CSF, IFN-γ, TNF-α, IL-2, IL-6, or bacterial lipopolysaccharides (LPS) [38,39,40,41,42]. Hsieh et al. [43] have elucidated the molecular basis for spontaneous PMN apoptosis. The group found that both Fas and Fas ligand (FasL) molecules are simultaneously expressed on the PMN surface. In addition, other apoptosis-related molecules, *p53* and *BCL-2* but not *c-myc*, are also expressed in PMN cytoplasm. Interestingly, the FasL molecules were found rapidly disappearing after 24 h of incubation. The authors concluded that the Fas-mediated pathway after Fas–FasL interaction is one of the molecular mechanisms in inducing spontaneous PMN apoptosis after PMN–PMN interactions. It has also been found that PMN apoptosis is tightly regulated by a complex network of signaling pathways controlling the key protein molecules of the *BCL-2* family via activation of MAP kinases, NF-κB, and caspase-degraded *BCL-2* homolog, as well as Myeloid Cell Leukemia-1 (MCL-1) [44,45]. MCL-1, as a survival molecule [46,47], can sustain PMN survival via heterodimerization with and neutralization of proapoptotic *BCL-2* family members, Bim or Bak, in the mitochondrial outer membrane [48,49,50]. On the contrary, the FasL, by bridging extracellular domains of TNF or TRAIL to membrane “death receptors”, TNF-R1 or TRAIL-R1/R2, can activate cytoplasmic “death domains”, FADD or FADD/TRADD. The activation then triggers caspase-8 and -3 to induce apoptosis [51,52]. It is worthy to note that the influence of microbes in alteration of the evolving routes of PMN is highly variable [53]. It could be microbe-specific, ranging from prolongation of PMN lifespan to rapid PMN breakdown after microbe phagocytosis. The molecular basis of the factors implicated in the differentiation of PMNs is shown in Figure 3.

## 3. Novel Biological/Immunological Functions of PMNs

PMNs are traditionally regarded as the first-line defending cells against microbial invasion by the way of phagocytosis, intracellular proteolytic killing and eradication of the microbes by reactive oxygen species (ROS). Nevertheless, a complete deletion of PMN (<0.5%) in rats with monoclonal anti-granulocyte antibody RP-3 that did not deplete innate and adaptive immune-related cells could alter the adaptive immune responses [54,55,56]. Yue et al. [57] and Dallegre et al. [58], in their in vitro studies, demonstrated that PMNs could exert cytotoxic effect in the presence of mitogen via a mitogen-induced cell-mediated cytotoxicity (MICC). Besides, other investigators discovered that antibody-dependent cell-mediated cytotoxicity (ADCC) is a universal immune activity mediated by IgG-Fc receptor-bearing cells including T cells, B cells, monocytes/macrophages and PMNs [59]. These results indicate that PMNs can actively participate in the effective immune responses in the body beyond the pathogen engulfing and killing functions. In this section, we will discuss more novel biological functions of PMNs involved in the immune network and immune homeostasis. Table 1 lists these new biological and immunological functions of PMN.

### 3.1. Biosynthesis and Secretion of Complement Component 3 (C3) and Factor B

Okuda T [60] firstly discovered that murine PMN can synthesize complement component C3 and factor B. Botto et al. [61] and Yu et al. [62] have subsequently confirmed that human PMN can synthesize and secrete functional C3 either spontaneously [62] or after activation by LPS or TNF-ɑ [61]. This PMN function would become an important mechanism for host defense at sites of inflammation in addition to granule proteins and ROS generation.

### 3.2. Release of Granule Proteins, Cytokines, Chemokines, and Growth Factors from PMNs for Cell–Cell Communication and Immune Modulation

PMNs not only play a pivotal role in the inflammatory reaction, but act as an important modulator in immune network by releasing a number of mediators. These mediators include granule proteins released through degranulation, and cytokines/chemokines/growth factors to actively take part in the immune system. The detailed steps of these functions will be discussed in the following sections.

#### 3.2.1. Degranulation to Liberate Azurophilic and Specific Granules

After activations by different stimuli, granule contents are released into the intracellular phagosomes or out to the extracellular space. This is called degranulation. During this process, membrane-attached granules fuse with phagosomes and plasma membrane, allowing a new biomarker expression on the cell surface [63]. Usually, the neutrophil granule proteins can be classified into azurophilic (primary) and specific (secondary) proteins. It should be noted that an excess of granule proteins release can lead to tissue damage, chronic inflammation and immune dysfunction [64,104]. Li et al. [65] found that the surface-expressed lactoferrin on PMN, a specific granule protein, could modulate Th1/Th2 cytokine production. In contrast, myeloperoxidase, an azurophilic granule protein, can induce chronic inflammation and NET formation [64].

#### 3.2.2. The Production of Cytokines/Chemokines/Growth Factors from PMNs for Immune Modulation

Bazzoni et al. [66] have firstly demonstrated that the phagocytosis of IgG opsonized-yeast particles by human PMNs can result in the expression and release of TNF-ɑ, but not IL-6. LPS stimulation on PMNs also releases TNF-ɑ. Since then, a bunch of cytokines/chemokines/growth factors have been successively discovered to be also produced and emancipated by PMNs. Compared to that produced by other innate or adaptive immune cells, the amount of cytokine production by individual PMN is relatively low. Nevertheless, this drawback can be partially compensated by the huge number of PMNs in the blood. In conjunction with the secreted cytokines/chemokines/growth factors [67,68,69,70,71] and the released granule proteins, NET formation as well as trogocytosis can render PMNs to intimately interact with the other immune-related cells, contributing essentially to the homeostasis of the immune system [105,106,107,108,109,110]. Table 2 summarizes the cytokine/chemokine/growth factor expressions in PMNs in vitro and in vivo, which is adapted from those reported by Cassatella et al. and Tsai et al. [110] with permission. 

#### 3.2.3. Liberation of Ectosomes and Exosomes from PMNs to Affect the Biological Functions of the Remote Cells or Tissues

In addition to the release of granule proteins/cytokines/chemokines/growth factors as mentioned in the Section 3.2.1, PMNs can also free extracellular vesicle (EV) to affect the biology/physiology of remote cells or tissues. The EVs released from PMNs contain two microvesicles, ectosomes (Ect) and exosomes (Exo), with a vesicle size ranging from 50 to 1000 nm in diameter. These PMN-derived EVs exhibit many important characteristics in the body defense, inflammatory responses and wound healing [72,73,74].

Stein et al. [111] have found that exocytosis of human PMNs can be elicited in the presence of a small amount of autologous complements. Later, Hess et al. [112] observed that N-formyl-methionyl-leucyl-phenylalanine (fMLP) or C5a can induce ectosome release from PMNs within a few minutes after stimulation. These extruded ectosomes contain a selective set of proteins originating both from the cell membrane and from the intracellular granule molecules such as neutrophil elastase (NE), myeloperoxidase (MPO), and proteinase 3 (Pr3) packed in the plasma membrane. However, a unique property of ectosomes is the exposure of phosphatidylserine (PS) in the outer leaflet of membrane [112], which is different from that of exosomes. In contrast, exosomes are defined as small membrane vesicles formed by inward budding of the endosomal membrane with little PS expression [113]. Functionally, ectosomes exhibit generic functions to downregulate inflammation and immunity, whereas exosomes potentiate the immune responses [114].

##### Suppressive Effects and the Signaling Pathways of PMN-Derived Ectosomes (PMN-Ect) on Macrophage Maturation

Gasser et al. [73] have shown that PMN-Ect could suppress the release of TNF-α, IL-8, and IL-10 from activated human monocyte-derived macrophages. Eken et al. [115] further disclosed that PMN-Ect could interfere with the LPS-stimulated dendritic cells maturation including morphological changes, phagocytic activity, surface molecule expression, cytokine release, and the capacity to induce T-cell proliferation. The same group then elucidated the signaling pathways of PMN-Ect-induced events against inflammation in macrophages/dendritic cells. These inhibitory processes induced by PMN-Ect include the MerTK pathway, Ca^2+^-flux and the release of stored TGF-β1 in the macrophages [116,117]. Expectedly, these PMN-derived granule cargos were also found to pave the way for tumor growth and progression [118], an untoward effect for the host.

##### The Modulatory Roles of PMN-Derived Exosomes (PMN-Exo) on the Immune Responses

Exosomes are the endosomal-derived microvesicles with 30–100 nm in diameter. These vesicles contain lipids, proteins and nucleic acids (DNA, mRNA and microRNA [miRNA]) inside. Many normal cells including immune-related cells (T cells, B cells, dendritic cells, mast cells and PMNs), neuron cells, astrocytes, epithelial cells and various tumor cells can release Exo. Once Exo are extruded into the extracellular milieu, they can be found in many biological fluids including plasma, urine, effusion, synovial fluid, saliva, or breast milk [119]. These microvesicles can transfer their cargo into the remote recipient cells to modify their biological activities. 

It is worthy to note that miRNAs in the Exo can modify the biological processes of the recipient cells, leading to autoimmune diseases [120,121] including SLE [122,123], RA [124], and systemic sclerosis (SSc) [125]. In particular, PMN-Exo can be involved in different pathological processes [126] in SSc [127] and dermatomyositis [128] via deranging various signaling pathways. These pieces of evidence demonstrate again that PMNs act as a double-edged sword to exert paradoxical effects in clinical medicine.

### 3.3. Induction of MHC-II Expression on PMNs by T-Cell-Derived Cytokines, Rendering PMN Mimicking Antigen-Presenting Cells (APC)

PMNs express a wide range of pattern recognition receptors [29], IgG-Fc receptors, and complement receptors [2] in resting state for mediating innate immune responses, as mentioned in Section 2.2. Interestingly, Oehler et al. [129] demonstrated that 9-day culture of PMNs in the presence of GM-CSF, IL-4 and IFN-α can augment the allogeneic stimulatory activity of the specific tetanus toxoid antigen after mixing with autologous memory T cell. Later, many authors found that PMNs activated by the memory T-cell-derived cytokines can express MHC-II and co-stimulatory molecules of CD80 and CD86, behaving like antigen-presenting cells [75,76,130]. Vono et al. [77] found that the cognate antigens of cytomegalovirus pp65- or influenza hemagglutinin-pulsed PMNs can present these two antigens to the autologous antigen-specific CD4^+^T cells in an MHC-II-dependent manner. These data support that PMNs acquire the capacity for antigen presentation to memory type CD4^+^ T cells in vitro and ex vivo. In addition, Meinderts et al. [78] demonstrated that human PMNs acquired antigen-presenting phenotype with expression of MHC class-II and co-stimulatory molecules, CD40 and CD80, following engulfment of IgG-opsonized erythrocytes. Besides, Polak et al. [79,80], by investigating IgE-mediated allergy, found that HLA-DR(+)-PMNs, and allergen-specific T cells accumulate in the sites of allergic late-phase reaction (LPR). In an in vitro experiment, the group further found that in the presence of a cocktail of GM-CSF, IFN-γ, and IL-3, PMNs internalized, processed, and then presented the allergen via HLA-DR loci to elicit the proliferation and cytokine production of the allergen-specific T cells. These results can support the potential antigen-processing capacity of PMNs in the stimulation of T cells or in the production of T-cell-derived cytokines. 

### 3.4. Trogocytosis (Plasma Membrane Transfer) among PMN, Non-Immune, and Immune-Related Cells

Trogocytosis (trogo means gnaw) is characterized by the transfer of plasma membrane fragments between two cells in contact after forming an immunological synapse. It has been demonstrated that trogocytosis is an active energy-consuming rapid transfer process after a conjugation between two homogeneous or heterogeneous living cells. The energy-requiring processes in trogocytosis include actin polymerization, membrane remodeling, signaling transfer, and finally the plasma membrane merges in the cell surface. This kind of membrane transfer has been demonstrated among immune, non-immune and even microbial cells [81]. The biological significance of trogocytosis may include: (1) cell–cell information exchange, (2) growth during embryonic development, (3) “nibbled to death” of infectious microbes, (4) immunoregulation, and (5) cancer immunity [82]. The following sections will discuss in detail the biological significance of trogocytosis mediated by PMNs.

#### 3.4.1. Elimination of the Intracellular Parasites or Unwanted Cells by PMN-Mediated Trogocytosis

Mercer et al. [131] have found that human PMNs can kill *Trichomonas vaginalis* in a dose-dependent, contact-dependent, and NET-independent manner via “bites” of the parasites until death. Both trogocytosis and parasite killing are dependent on the presence of PMN’s serine proteinase and human serum factors. Furthermore, Olivera-Valle et al. [132] found that PMNs attacked and killed excessive exogenous immobile sperms in the vagina via trogocytosis with high efficiency after contact with these sperms without inducing vaginal mucosa damage or infertility.

Taylor et al. [83] are the first authors to propose a specialized form of trogocytosis mediated by Fcγ receptors (FcγR) on effector cells in cancer immunotherapy by using anticancer monoclonal antibodies. The hypothesis is further supported by Valgardsdottir et al. [84] that PMNs can carry out mostly trogocytosis rather than phagocytosis of the anti-CD20-opsonized chronic lymphocytic leukemia cells in autoantibody-based anticancer therapies.

#### 3.4.2. Trogocytosis among PMNs and Other Immune-Related Cells for Immune Modulation

Poupot et al. [85] have found that spontaneous membrane transfer occurs among homotypical leukemia cell lines without stimulation of the prolongation of cell survival. Honer et al. [86] demonstrated the occurrence of trogocytosis between PMNs and tumor cells in the presence of antitumor antibodies. Li et al. [87] have observed many differences in the mechanisms and biological significance of trogocytosis between normal human PMNs and mononuclear cells (MNCs). The group disclosed that membrane transfer from MNCs to PMNs occurred at the site of immunological synapse for transducing survival and activation signals. The membrane transfer from MNCs to PMNs enhances PMN functions, which is dependent on actin polymerization, clathrin activation and the presence of Fcγ receptors. On the other hand, membrane transfer from PMNs to MNCs depends on MAP kinase and PKC signaling pathways for cell–cell communication and immune modulation.

To date, trogocytosis has been found among various types of immune cells including T cells [88], B cells [89], NK cells [133], macrophages/DCs [134], basophils [135], and innate lymphoid cells [136]. In a biological sense, trogocytosis participates in the elimination of invading pathogens and tumor cells to protect the body. On the other hand, in an immunological sense, trogocytosis is implicated in the activation or inhibition for the immune homeostasis [137,138,139].

### 3.5. Biological and Pathobiological Roles of NET Formation from PMNs

The most powerful antimicrobial mechanism by PMNs is the extrusion of the intracellular structures in the form of NETs into the surrounding environment [140]. NETs can not only effectively trap the invading pathogens for preventing their spread, but also rapidly kill the pathogens by the chromatin-attached granule molecules such as proteinase, elastase, myeloperoxidase and LL-37.

NETs are large web-like structures composed of cytosolic and granule proteins assembled on a scaphoid of decondensed chromatins derived mostly from nucleus and less from mitochondria. However, once the clearance of this exposed web-like structure in the blood vessels or tissues is insufficient, NETs may induce immune-mediated diseases such as rheumatoid arthritis [141]. This is because DNA-containing NETs per se can stimulate proinflammatory cytokine production via TLR9 on the innate immune cells [90]. In a clinical sense, the excessive NET formation in the blood vessels may induce a wide range of pathological conditions including vascular thrombosis/atherosclerosis [142,143,144], autoimmune diseases [145,146] and tumor progression/metastasis [143,147]. Conversely, impaired NET formation may be found in aged individuals due to defective innate immunity [148]. Recently, Lu et al. [149] found that FcγRIII engagement can augment PMA-stimulated NET formation partially via cross talk between Syk-ERK-NF-κB and PKC-ROS signaling pathways. Farrera et al. [150] reported that the clearance of NETs depends on complement components C1q, DNase 1, C-reactive protein and macrophage engulfment. 

## 4. Heterogeneity of PMN in Facilitating or Deterring Tumorigenesis

Evidence has shown that the functional heterogeneity of PMNs depends on the tumor microenvironment (TME) for determining either pro- or antitumor effect. PMNs exhibit an antagonizing effect at the early stage of tumorigenesis [151], whereas they exhibit a facilitating effect [11] at the late stage of tumorigenesis. It has been reported that G-CSF [152] and IL-8 [153] are produced from tumor cells and tumor-surrounding cells for skewing the number and nature of neutrophils in the TME. Thereby, tumor-associated PMNs (TANs) contain heterogeneous populations with different functional capacities ranging from effector to myeloid suppressor cells [154,155]. Arbitrarily, the TANs can be classified into antitumor neutrophils (N1) upon stimulation by IFN-β [156] and tumor-promoting neutrophils (N2) upon stimulation by TGF-β [157]. The antitumor N1 are characterized by high expression levels of TNF-α, Fas, and ICAM, but low expression level of arginase [158]. In addition, N1 cells can release ROS, proteolytic enzymes, NETs and exhibit high adherence capacity to destroy the tumor cells. They can also promote T-cell immunity by recruitment and activation of CD8^+^ T cells via IFN-α production to induce cell-mediated cytotoxicity [159]. 

The possible mechanisms of N1 cells to kill tumor cells are through (1) ADCC [160], (2) antibody-opsonized FcγR-mediated trogocytosis [161], and (3) FcαR-mediated antibody-dependent ADCC that can be potentiated by CD47-SIRPA checkpoint blockade [161,162]. On the contrary, FcγRIIIb (as a decoy receptor), which is different from FcγRIIa, may restrict antibody-dependent destruction of cancer cells by human PMNs [163]. Recently, NETs have been found to exert the pro-tumor effects on the malignant tumor progression via mechanisms involving the establishment of an inflammatory microenvironment in association with other pro-tumor mechanisms such as inflammasomes or autophagy [164,165,166,167]. The potential antitumor mechanisms by N1 neutrophils are illustrated in Figure 4.

In addition, a particular group of granulocytic myeloid-derived suppressor cells (MDSCs) is widely described as an immature subset by their ability to impede both innate and adaptive immunity. Accordingly, this subset of cells have been demonstrated both in vitro and in vivo to inhibit T-cell immunity as well as to promote the growth and spread of cancers [168,169]. The neutrophil heterogeneity in facilitating and deterring tumorigenesis is summarized in Table 3. 

## 5. Impact of PMNs on Cardiovascular Disease (CVD)

Besides the role of PMN-NETs on vascular thrombosis and atherosclerosis [142,143,144], as mentioned in Section 3.5, inflammation is also regarded as a risk factor for CVD in apparently healthy people [170], independent of dyslipidemia [171]. PMNs have been found to destroy the endothelial cells (ECs) by releasing Pr3, NE [172] and MPO [173]. These proteolytic enzymes can break down basement membranes and induce endothelial apoptosis. Indeed, the serum levels of Pr3 and NE are found to increase in patients with acute myocardial infarction (AMI) [174,175]. In addition, release of ROS from PMN is another potent EC-damage factor in CVD [176]. Recently, Sreejit et al. [177] discovered that a PMN-derived alarmin (i.e., S100A8/A9) can induce inflammation and cardiac injury after myocardial infarction. Critical reviews for the impact of CVDs on PMNs have been reported by Bonaventura et al. [178] and Silvertre-Roig et al. [179].

## 6. The Role of PMNs in Antiviral Infection Processes

Douglas et al. [91] firstly noted that certain viruses invading the respiratory system can induce hyper-neutrophilia and neutrophil responses. Faden et al. [92] have demonstrated that PMNs can adhere to the respiratory syncytial virus-infected cells. Furthermore, MacGrigor et al. [93] and Ratchiffe et al. [94] have found that PMNs can interact with virus-infected cells to increase granulocyte adhesion. Thus, viral infections may provide the necessary stimuli for PMN migration, adhesion, and responses. In animal experiments, Rouse et al. [95] found that the PMNs collected from bovine mammary glands in co-culture with infectious bovine rhinotracheitis (IBR) virus-infected Georgia bovine kidney cells could potently induce type 1 interferon. Tumpey et al. [96] further demonstrated that Herpes simplex virus type 1 (HSV-1)-infected murine cornea could induce rapid infiltration of PMNs in both immunocompetent and immunodeficient mice. These infiltrated murine PMNs can suppress virus replication and spread after the corneal infection. Besides, murine PMNs have been proven capable of limiting influenza virus [97,98] and HSV-1 [99] spreading in mouse models. To explore the molecular basis for antiviral activity of PMNs, Tate et al. [100] investigated the impact of PMNs on CD8^+^ T-cell responses in the virus-infected airway and secondary lymphoid tissues. The authors concluded that PMNs can sustain effective CD8^+^ T-cell responses in the influenza virus-infected respiratory tracts in mice.

The defense mechanisms adopted by PMNs in antiviral immunity have been studied by many authors, which include phagocytosis, degranulation, induction of respiratory burst, secretion of cytokines/chemokines, and release of NETs [101,102]. Hayashi et al. [103] demonstrated that PMNs expressed a broad spectrum of pathogen recognition receptors (PRRs) on the cell surface for recognition of PAMPs in viruses to induce immune responses. TLRs are considered classical PRRs that can detect viral proteins and nucleic acids. TLR3 recognizes double-stranded RNA, TLR7 and TLR8 recognize single-stranded viral RNA, whereas TLR9 recognizes unmethylated CpG DNA of the viruses [103]. Furthermore, Gan et al. [180] unveiled that TLR3 regulated poly I:C-induced neutrophil NETs and elicited acute lung injury through p38 MAP kinase. Besides, Stegelmeier et al. [181] proved that type I interferon can regulate the antiviral capacities of PMNs. In short conclusion, the complex cross talk between PMNs and adaptive immune cells via type I IFNs and TLRs can exert antiviral effects in PMNs. 

## 7. Overwhelming Immune Responses Relevant to PMNs in Pandemic Coronavirus Disease

Coronavirus disease 2019 (COVID-19 or SARS-CoV-II) is a virus-infected respiratory disease that can potentially progress to acute respiratory distress syndrome (ARDS), cytokine storm, and multiple-organ failure. This hyper-inflammatory status is originated from dysregulatory immune responses as a result of impaired T-cell suppression and excessive innate immune activation [182,183,184]. Some of the patients with COVID-19 infection displayed severe lymphopenia (decrease in CD4^+^ T, CD8^+^ T, and B cells) and delayed lymphocyte activation [185]. In contrast, elevated serum concentration of IP-10 (IFN-γ-induced protein 10) and GM-CSF in COVID-19 disease is compatible with enhanced T-cell and monocyte recruitments, increased proinflammatory cytokine production and excessive PMN chemotaxis [185]. Accordingly, the overwhelming innate and adaptive immune responses inevitably elicit ARDS in some patients with COVID-19 disease [186]. Yao et al. [187], by analyzing transcriptomic profiles of immune-related cells, revealed that defective antigen presenting and IFN responsiveness of monocytes were present in patients with COVID-19-induced ARDS, in contrast to higher lymphocyte responsiveness. Besides, the gene expression involved in cytotoxic activity was decreased in NK, CD8^+^T and B cells in these patients. Delayed viral clearance is also found in severely ill COVID-19 patients. de Candia et al. [188] found that the mortality of SARS-CoV-2 infection was higher in the elderly than in younger adults and apparently rare in children, which seemed attributable to the innate and adaptive immune status. Although PMNs can facilitate virus clearance as mentioned in Section 6, neutrophilic inflammation may conversely contribute to the higher mortality of COVID-19 patients with underlying comorbidities such as diabetes or CVD. The molecular mechanisms of PMNs in triggering severe complications such as cytokine storm, hyperinflammation/necroinflammation, immunothrombotic dysregulation, and multi-organ failure in relation to NETs formation in COVID-19 disease will be discussed in the following sections. 

### 7.1. The role of PMN-Derived NETs in Inducing Hyperinflammation, Lung Cell Death, Cytokine Storm, ARDS, and Immunothrombotic Dysregulation in COVID-19 Disease

Scientists have observed that an increased number of circulating PMNs may become an indicator for the severity of respiratory symptoms and poor prognosis in COVID-19 infection [189]. PMN-derived NETs are considered one of the potent inducers in lung inflammation. Veras et al. [190] unraveled that viable SARS-CoV-2 can trigger NETs release from PMNs, which is dependent on the activation of angiotensin-converting enzyme receptor 2 (ACE-R2), serine proteinase, and peptidylarginine deiminase 4 (PAD4) in addition to viral replication per se. The activation of these molecules can be reflected by elevated NET concentration in plasma, tracheal aspirate and autopsied lung tissues in severely ill patients. Moreover, these NETs derived from PMNs can concomitantly cause death of the pulmonary epithelial cells and hyperinflammation of the lungs [190,191,192,193].

Cytokine storms are usually driven by the unrestrained activation of leukocytes to produce a huge amount of proinflammatory cytokines including IL-1β, IL-6, TNF-α, IL-17, IFN-α, along with PMN-NETs release. These pathological factors elicit severe damage on the inflamed blood vessels and infiltrated tissues, especially in patients with diabetes and hypertension [194,195,196].

### 7.2. The Interactions of NETs, Complements, Coagulation Factors, and Platelets in the Immunothrombotic Dysregulation in COVID-19 Infection

Nicolai et al. [197] examined the autopsied cases of COVID-19 disease and found extensive inflammatory microvascular thrombus formation in the lung, kidney and heart tissues containing NETs in association with platelets and fibrin. Besides, patients with COVID-19 disease also exhibited PMN-platelet aggregates with distinct PMN and platelet activation markers in blood. Middleton et al. [198] have also confirmed NET-containing microthrombi with PMN and platelet infiltration in autopsied lungs. Radermecker et al. [199] disclosed that NETs could contribute to inflammation-associated lung damage, thrombosis, and fibrosis in severely ill patients. Regarding the molecular basis of NET-induced immunothrombus formation in COVID-19-infected patients, Skendros et al. [200] found increased plasma NETs, tissue factor (TF) activity, and soluble complements sC5b-9 in COVID-19 patients. In addition, PMNs from those patients released high amounts of TF and NET-carrying active TF. These results have indicated a pivotal role of complements and NETs in immunothrombus formation in COVID-19 patients. Busch et al. [201] have explored the role of intrinsic pathway of coagulation cascades including plasma kallikrein and bradykinin in COVID-19-induced immune thrombosis. They concluded that the hypercoagulability and thrombotic events in COVID-19 patients are driven by PMN-NETs, complement activation, and contact activation. The use of C5a blockers, the plasma kallikrein and blood coagulation factor XIa (FXIa) inhibitors, and agents neutralizing extracellular histones may be beneficial for the control of COVID-19-induced immunothrombotic dysregulation.

Recently, Ercan et al. [202] analyzed phenotypic changes of platelets in COVID-19 patients and found a decrease in the total amount of integrin αIIb (ITGA2B), a subunit of αIIbβ3, in the patients compared to healthy controls. Higher consumption of fibrin-stabilizing factor, i.e., coagulation factor XIIIA (F13A1), in platelets was found in COVID-19 patients. Conversely, increased amounts of annexin A5, eukaryotic initiation factor 4A-1 (ElF4A1) and transaldolase 1 (TALDO1) in platelets were correlated to nasopharyngeal COVID-19 viral load. Furthermore, the levels of 2 protein disulfide isomerase, P4HB and PD1A6, which facilitate thrombosis, were found to be increased in COVID-19 disease. The crucial role of PMNs in the immunothrombosis in COVID-19-infected patients has been critically reviewed by Iliadi et al. [203] and Bautista-Becerril et al. [204].

A proposed molecular basis for PMN-derived NETs in the pathogenesis of lung cell death, hyperinflammation, cytokine storm, multiple-organ failure, vascular damage, and immunothrombotic dysregulation in SARS-CoV-infected patients is provided in Figure 5. 

## 8. Conclusions

A number of novel biological/immunological functions of PMNs have been successively found. Some of the unique functions including NET formation, N1/N2 neutrophil and granulocytic MDSCs heterogeneity, MHC-II antigen expression, trogocytosis, extrusion of ectosomes and exosomes, and antiviral activity can act as a fast and effective defender to protect the body. On the contrary, PMNs may also play a dual role to exhibit paradoxical activities to favor or to oppose inflammation, to exert antimicrobial effect or autoimmunity, to exhibit pro-cancer or anticancer behaviors, to present antiviral effect or facilitate SARS-CoV-II-induced immunothrombotic dysregulation in clinical medicine. The NETs derived from activated PMNs play a pivotal role in these paradoxical activities. It is quite important to further elucidate more molecular evidence for this strange phenomenon and to design an effective therapeutic strategy for overcoming these untoward effects of PMNs in clinical medicine as shown in Table 4. Accordingly, we propose several tips to take advantages of these paradoxical characteristics of PMNs for SARS-CoV-2 therapy, as shown below:(1)Adequate use of the PMN-released defensins (HNP1, HD5) or antimicrobial peptide retrocyclin-101 (RC101) to block SARS-CoV-2 entry.(2)Facilitating NET release initially to trap and kill the virus followed by adding of complement C1q to rapidly clear the accumulated NETs.(3)Application of monoclonal anti-IL-17 antibody to suppress SARS-CoV-2 virus-induced hyperinflammation.(4)Induction of granulocytic MDSCs by TGF-β for anti-inflammation effect and immunosuppression of the cytokine storm.

## Figures and Tables

**Figure 1 biomedicines-10-00773-f001:**
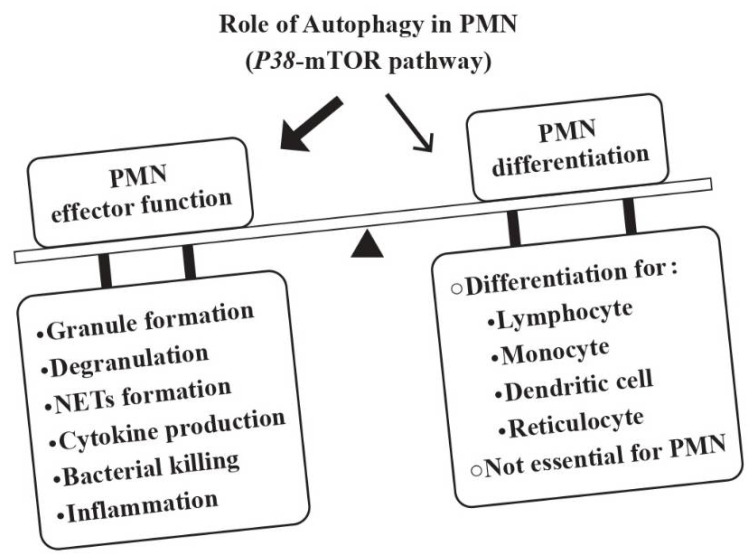
The roles of autophagy in the development and effector functions of PMNs. Although autophagy is essential for the ontogenetic development of lymphocytes, monocytes/dendritic cells and reticulocytes, it is not essential for neutrophils. Instead, the *p*38-mTOR-induced autophagy is pivotal for developing effector functions of PMNs including granule formation, degranulation, NETs formation, cytokine production, microbial killing, and inflammation.

**Figure 2 biomedicines-10-00773-f002:**
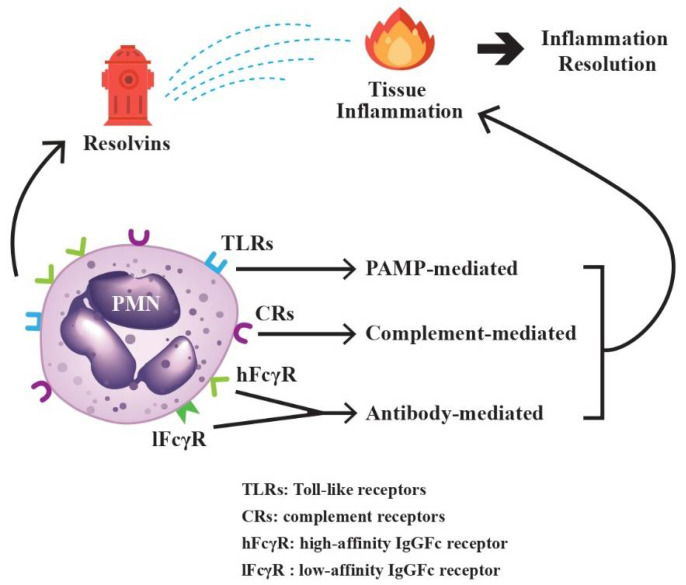
The paradoxical activities of PMNs on tissue inflammation and inflammation resolution. The surface-expressed Toll-like receptors (TLRs) bind to pathogen-associated molecular patterns (PAMPs), complement receptors (CRs) bind to antigen-antibody-complement immune complexes, and high-affinity (hFcγR) and low-affinity IgG Fc receptors (lFcγR) bind to IgG antibodies, which then can induce tissue inflammation. On the other hand, resolvins, the metabolic products of omega-3 polyunsaturated fatty acid synthesized during the initial phases of acute inflammatory responses, can promote the resolution of inflammation. Both inflammation and its resolution can be mediated by PMNs for immune homeostasis.

**Figure 3 biomedicines-10-00773-f003:**
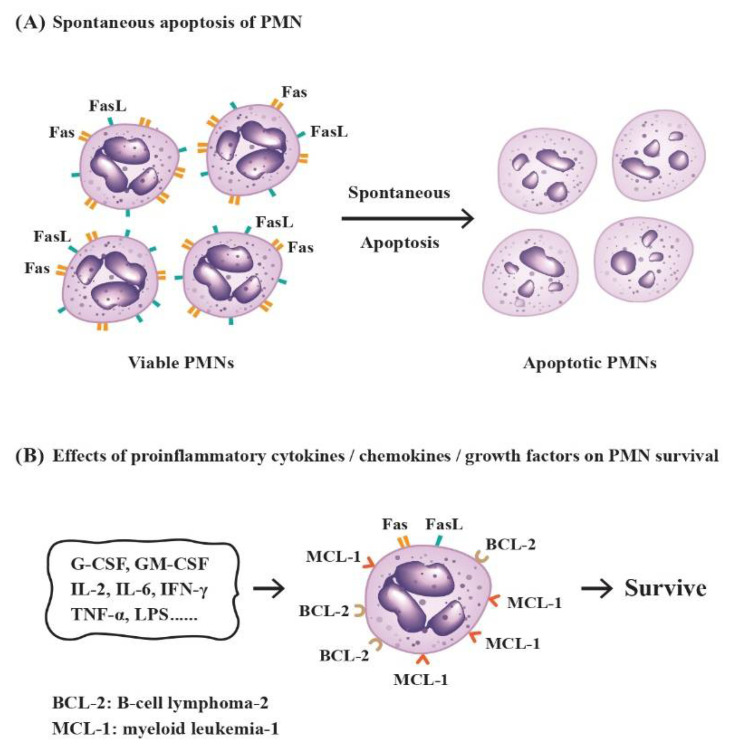
The molecular basis for spontaneous apoptosis and survival prolongation of PMNs by proinflammatory cytokines/chemokines/growth factors in the physiological or inflammatory environment. (**A**) Induction of spontaneous PMN apoptosis by interactions of Fas ligand (FasL) and Fas receptor (Fas, CD95) expressed on the cell surface of neighboring PMNs in normal condition; (**B**) The lifespan of PMNs can be prolonged by inflammation-related factors in the environment via increased expression of survival molecules BCL-2 and MCL-1.

**Figure 4 biomedicines-10-00773-f004:**
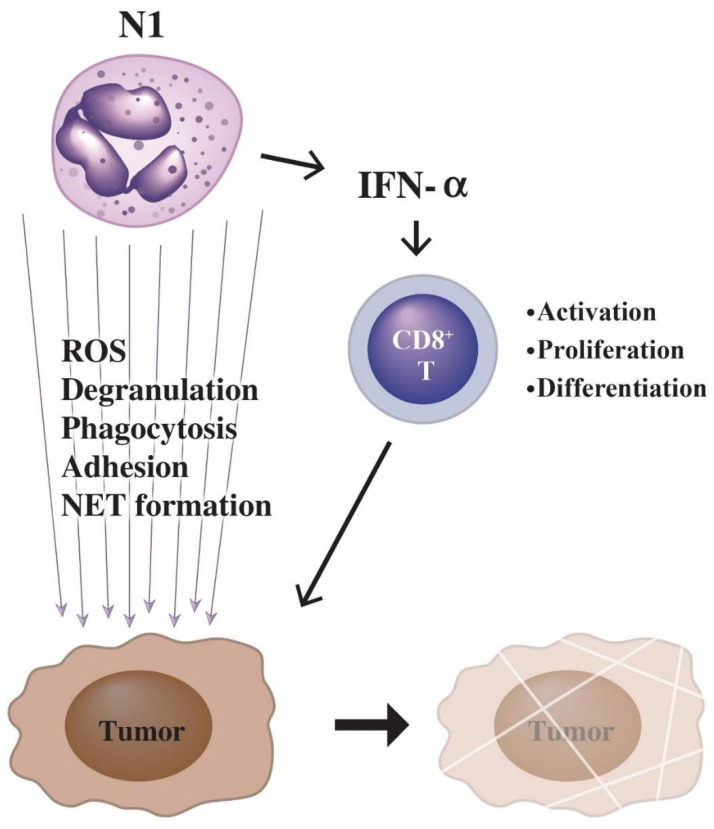
Killing of tumor cells by N1-type PMNs is through two pathways: (1) enhanced tumor-killing activity of cytotoxic CD8^+^ T-cell by IFN-α released from PMNs; (2) tumor-killing molecules released from PMNs including ROS, granule proteins, and NETs in association with phagocytosis and adhesion of tumor cells by PMNs.

**Figure 5 biomedicines-10-00773-f005:**
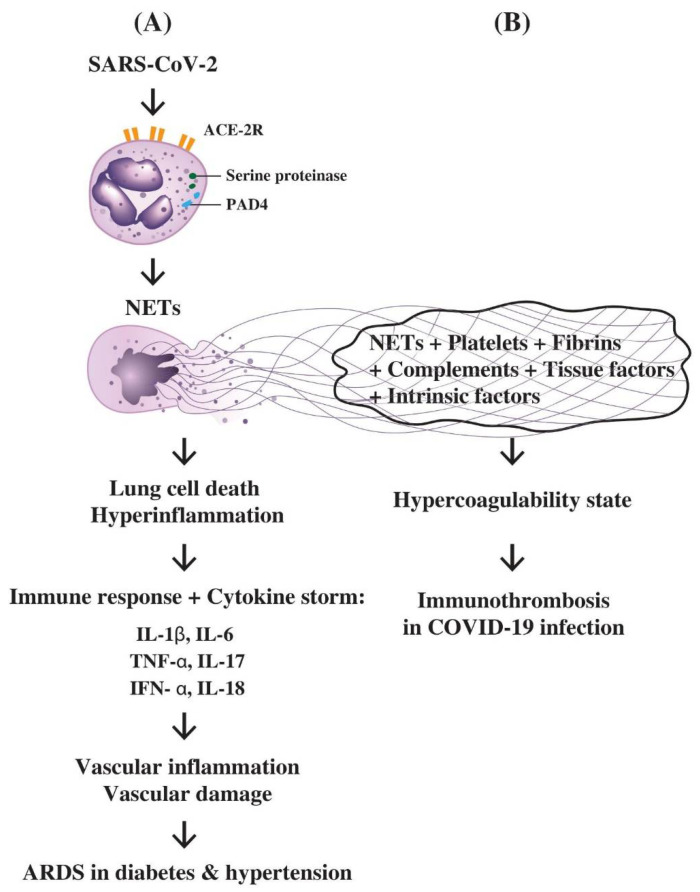
The molecular basis of PMNs in inducing hyperinflammation of lung, cytokine storm and immunothrombosis during SARS-CoV-2 infection in patients with diabetes or hypertension. (**A**) Attachment of SARS-CoV-2 virus to ACE2 receptors on PMN results in the enzymatic activation of serine proteinase and PAD4 to induce NETs formation. The NETs can cause pulmonary epithelial cell death and hyperinflammation in the lung. The overwhelming immune responses with profound proinflammatory cytokine production including IL-1β, IL-6, IL-8, IL-17, IFN-α, and TNF-α elicit cytokine storm, extensive vascular inflammation and damage leading to acute respiratory distress syndrome (ARDS); (**B**) The trapping of platelets, fibrins, complements, tissue factors, intrinsic coagulation factors, and tissue debris by NETs released from PMNs lead to a hypercoagulability state of the body. Finally, immunothrombosis in the lung, heart, brain, kidney or gastrointestinal tract occurs in COVID-19 disease.

**Table 1 biomedicines-10-00773-t001:** Novel biological/immunological functions of PMN.

Functions	References
Mitogen-induced cell-mediated cytotoxicity (MICC)	[57,58]
Antibody-dependent cell-mediated cytotoxicity (ADCC)	[59]
Biosynthesis & secretion of complement components C3 and factor B	[60,61,62]
Degranulation of azurophilic and specific granule proteins	[63,64,65]
Production of cytokines/chemokines/growth factors	[66,67,68,69,70,71]
Exocytosis of ectosomes and exosomes	[72,73,74]
Expression of MHC-II antigens for antigen-presenting activity	[75,76,77,78,79,80,81,82]
Trogocytosis (plasma membrane exchange) by PMN	[83,84,85,86,87,88,89]
Neutrophil extracellular traps (NET) formation by PMN	[90]
Antiviral activity by PMNs	[91,92,93,94,95,96,97,98,99,100,101,102,103]

**Table 2 biomedicines-10-00773-t002:** The cytokines/chemokines/growth factors expression in normal human PMNs in vitro and in vivo *.

In Vitro	In Vivo
IL-1α/IL-1β	IL-1α
IL-1ra	IL-1β
IL-8	IL-1ra
IL-12	IL-6
TNF-α	IL-8
IFN-α	IL-10
CD30L	IL-12
GROα, GROβ	MIP2
CINC-1, 2a, 3	KC/GROα
IP-10	CINC
MIG	MIP-1α
MIP-1α/-1β	MIP-β
TGF-α, TGF-β1	MCP-1
IL-3, G-CSF, M-CSF	TNF-α
GM-CSF (?)	TGF-β1
IL-6 (?)	
MCP-1 (?)	
SCF (?)	

* Adapted from Cassatella, M.A. et al.’s “The neutrophils: new outlook for old cells”. Imperial College Press 1999, pp. 151–299 and Tsai et al. Clin Exp Rheumatol 2019, 37, 684–693. CINC-1: cytokine-induced neutrophil chemoattractant 1. MIG: monokine induced by interferon gamma = CXCL-9. SCF: stem cell factor. CSF: colony stimulating factor. IL-1ra: IL-1 receptor antagonist. MIP: macrophage inflammatory protein. MCP: macrophage chemotactic protein. (?): supporting evidence is insufficient.

**Table 3 biomedicines-10-00773-t003:** The functions and characterization of different heterogeneity of neutrophil subpopulations including N1, N2, and granulocytic MDSCs, relevant to the tumorigenesis.

	Neutrophil Type 1 (N1)	Neutrophil Type 2 (N2)	Granulocytic MDSC
Function	Anti-tumor	Pro-tumor	Pro-tumor
Stimulated by	IFN-β [156]	TGF-β [157]	Interferon regulatory factor-8 deficiency [168]
Expression of	TNF-α, Fas, ICAM	FcγRIIIb [163]	CD11b^+^ Ly6C^lo^Ly6G^+^
	ROS, NET		
	Proteolytic enzymes		
	Arginase [158]		
Immunity	Promote CD8^+^ T activation & cell-mediated cytotoxicity [160,161,162]	CD8^+^ T cells	Suppressive effect on T cell immunity [167,168]
		IL-4 and IL-13 secretion [157]	Suppression of NK activity
	ADCC [159]		
	FcγT-mediated trogocytosis [161]		
	FcαR-mediated ADCC [162,163]		

**Table 4 biomedicines-10-00773-t004:** Untoward effects of deranged PMN functions in clinical practice.

Untoward Effects	Pathology & References
PMN-derived ectosomes	Tumor growth and tumor progression [119]
PMN-derived exosomes	Systemic sclerosis [127,128] Dermatomyositis [129]
Excessive NETs formation or insufficient NETs clearance	Rheumatoid arthritis [90] Vascular thrombosis/atherosclerosis [143,144,145] Autoimmune diseases [146,147]
Impaired NETs formation by PMN	Tumor progression/metastasisInfection [144,148]
Excessive N2 in tumor-associated neutrophils oversecreting TGF-β and arginase	Tumorigenesis [158]
FcγRIIb on PMN	ADCC-mediated tumor cell killing [164]
Abnormal NET formation	Pro-tumor effects by inflammatory microenvironment Interaction with inflammasomes and autophagy
Release of proteinase 3, neutrophil elastase, and myeloperoxidase	Endothelial cell apoptosis Acute myocardial infarction [171,172,173,174]
Release of Alamins (S100A8/A9)	Inflammation and cardiac injury [171,172,173,174,175,176,177,178]

## Data Availability

Not applicable.

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
