# Peer review of "Molecular Basis for Paradoxical Activities of Polymorphonuclear Neutrophils in Inflammation/Anti-Inflammation, Bactericide/Autoimmunity, Pro-Cancer/Anticancer, and Antiviral Infection/SARS-CoV-II-Induced Immunothrombotic Dysregulation"

_biomedicines, 2022, doi:10.3390/biomedicines10040773_

Round 1

Reviewer 1 Report

Well written narrative review.

Please correct typos in paragraphs 3.5.1, 3.5.2. and 4

Recently it has become apparent that neutrophils have many crucial homeostatic functions in various organ systems, and their new roles are emerging. This article is a systematic review about the molecular basis of main neutrophil activities. It can not be considered "original" in terms to add new knowledge to the knowledge that already exists in this research area, as there are countless papers on emerging functions of neutrophils. The reference list has detailed sources cited in the text. 

Author Response

As attached in the separate answer sheet .

Reviewer 2 Report

It is a good and timely review.

I have some suggestions for improving it:

The authors should add a new table recapitulating the most important points regarding heterogeneity of PMN in facilitating vs. deterring tumorigenesis.

Please discuss, even as future perspectives, potential importance / differences in PMN functions and responses based on different SARS-CoV-2 variants.

Paragraph 6 should be implemented also using updated references and adding more data; now it looks too much “superficial”.

Author Response

As attached in the separate answer sheet.
